# Efficacy of Repetitive Transcranial Magnetic Stimulation (rTMS) Combined with Psychological Interventions: A Systematic Review and Meta-Analysis of Randomized Controlled Trials

**DOI:** 10.3390/brainsci13121665

**Published:** 2023-11-30

**Authors:** Xiaomin Xu, Mei Xu, Yon Su, Thanh Vinh Cao, Stevan Nikolin, Adriano Moffa, Colleen Loo, Donel Martin

**Affiliations:** 1Discipline of Psychiatry & Mental Health, School of Clinical Medicine, Faulty of Medicine and Health, University of New South Wales, Sydney, NSW 2052, Australia; 2Black Dog Institute, Hospital Road, Randwick, NSW 2031, Australia

**Keywords:** repetitive transcranial magnetic stimulation, psychological interventions, systematic review, meta-analysis

## Abstract

(1) Background: Psychological interventions are effective in alleviating neuropsychiatric symptoms, though results can vary between patients. Repetitive transcranial magnetic stimulation (rTMS) has been proven to improve clinical symptoms and cognition. It remains unclear whether rTMS can augment the efficacy of psychological interventions. (2) Methods: We examined the effects of rTMS combined with psychological interventions on clinical, functional, and cognitive outcomes from randomized controlled trials conducted in healthy and clinical populations. We searched PubMed, EMBASE, Cochrane Library, and PsycINFO databases up to April 2023. (3) Results: Twenty-seven studies were ultimately included. Compared to sham rTMS combined with psychological interventions, active rTMS combined with psychological interventions significantly improved overall clinical symptoms (k = 16, SMD  =  0.31, CIs 0.08 to 0.54, *p*  <  0.01). We found that 10 or more sessions of rTMS combined with cognitive behavioural therapy significantly improved clinical outcomes overall (k = 3, SMD  =  0.21, CIs  0.05 to 0.36, Z = 2.49, *p*  <  0.01). RTMS combined with cognitive training (CT) significantly improved cognition overall compared to sham rTMS combined with CT (k = 13, SMD  =  0.28, CIs 0.15 to 0.42, *p*  <  0.01), with a significant effect on global cognition (k = 11, SMD  =  0.45, CIs 0.21 to 0.68, *p*  < 0.01), but not on the other cognitive domains. (4) Conclusion: The current results provide preliminary support for the augmentation effects of active rTMS on clinical and cognitive outcomes across diverse populations. Future clinical trials are required to confirm these augmentation effects for specific psychological interventions in specific clinical populations.

## 1. Introduction

Psychological interventions, including cognitive behavioural therapy (CBT), mindfulness-based cognitive therapy, cognitive remediation, etc., aim to promote one’s ability to adapt to a given situation, leading to improved functioning [1]. Despite the increasing clinical use of psychological interventions, their efficacy can be limited. A systematic review of 419 randomized controlled trials (RCTs) from clinical and non-clinical populations found that psychological interventions produced only small-to-moderate effect sizes on mental well-being and/or indicators of illness [2]. Efficacy is often also lower in patients with more severe symptoms and in those who have failed other therapies [3]. The long duration (e.g., 3–12 months) required for a psychological intervention to take effect [4] is a further consideration. For these reasons, the combination of psychological therapies with biological therapies (e.g., pharmacotherapy) is often recommended in clinical practice guidelines [5,6]. While the efficacy of the combination of different psychological interventions with pharmacotherapies has been commonly studied [7,8], combining psychological therapies with non-invasive brain stimulation (NIBS) treatments is a relatively new area of research interest with the potential to improve therapeutic outcomes.

Transcranial magnetic stimulation (TMS) is a form of NIBS that has been extensively utilized for research and clinical applications [9,10]. TMS exerts its effects by delivering magnetic pulses that go through the scalp and skull, which induce electrical currents that can depolarize neurons in the brain [11,12]. Repetitive transcranial magnetic stimulation (rTMS), which delivers multiple repeated magnetic pulses, causes long-lasting effects on cortical excitability beyond the stimulation period [13]. RTMS can also improve cognitive functions in various domains in both healthy [14] and clinical populations [15,16,17]. RTMS parameters can be adjusted to have different effects: high frequency (≥5 Hz) rTMS and intermittent theta-burst stimulation (iTBS) facilitate neural activity, while low frequency (≤1 Hz) rTMS and continuous theta-burst stimulation (cTBS) inhibit it [9,18]. Theta burst stimulation (TBS), a newer form of rTMS, involves delivering patterned stimulation to the brain at specific frequencies (typically at a theta frequency with gamma frequency “bursts”) and has a shorter treatment duration than standard rTMS [19]. Evidence from several meta-analyses has shown that compared to sham stimulation, active rTMS can alleviate symptoms of depression [20] and improve clinical outcomes in other neuropsychiatric illnesses [21,22,23,24,25,26]. Research has shown that, typically, at least 20–30 rTMS sessions administered over consecutive weekdays are necessary for optimal clinical effects [27].

In vivo and in vitro studies have shown that rTMS can affect neurotransmission, influencing adaptive synaptic plasticity and producing long-lasting effects [24]. It can also prolong the time window for the neural interactions subserving behavioural adaptation [28]. Psychological interventions such as CBT and cognitive training (CT) can also facilitate synaptic plasticity [29]. Based on the above, it is possible that rTMS could potentially enhance the effectiveness of psychological interventions by maximizing plasticity. Neuroplasticity is considered to be a critical treatment target for clinical improvement and enhancing cognitive and functional abilities in various neuropsychiatric diseases [30,31]. Existing studies that have paired rTMS with psychological interventions, including exposure therapies [32], CT [33], and CBT [34] have provided preliminary evidence supporting the augmentation effect of rTMS on clinical outcomes across a broad range of neuropsychiatric disorders. While meta-analyses of RCTs of rTMS alone or psychological interventions alone for individual disorders are available, there is currently no systematic examination specifically focusing on the use of rTMS to augment different types of psychological interventions. 

In the current study, we aimed to investigate the potential augmentation effects of rTMS on psychological interventions in healthy and clinical populations for different outcomes (i.e., clinical, functional, and cognitive outcomes) to identify common trends and patterns. A secondary aim was to explore which type of psychological interventions may exhibit greater augmentation effects when combined with rTMS. To address these aims, we conducted a systematic review and meta-analysis of RCTs of active rTMS combined with psychological interventions compared to sham rTMS combined with psychological interventions. 

## 2. Materials and Methods

This systematic review and meta-analysis was conducted in accordance with the Preferred Reporting of Systematic Reviews and Meta-Analyses (PRISMA) statement and was registered on the International Prospective Register of Systematic Reviews, PROSPERO [35]. The PRISMA checklist for this systematic review and meta-analysis is provided in the Appendix A [36].

### 2.1. Search Strategy and Study Selection

Keywords (‘repetitive transcranial magnetic stimulation’ OR ’rTMS’, AND ’psychological interventions’ OR ’psychotherapies’, AND ‘randomized controlled trial’ OR ‘RCT’) were searched in four electronic databases including PubMed, EMBASE (Ovid), Cochrane Library, and PsycINFO (Ovid) up to 22 April 2023. The search strategies are shown in Appendix A. Studies were included if they fulfilled the following criteria: (1) healthy or clinical populations ≥ 18 years; (2) the comparison of rTMS combined with psychological interventions (e.g., CBT, exposure therapy, and CT) to sham rTMS combined with psychological interventions; (3) reporting pre-intervention and post-intervention assessment of clinical outcomes (i.e., symptom severity, functioning, and quality of life) or cognitive outcomes; (4) RCTs published in peer-reviewed English journals. After removing duplicates in the original search results, the titles and abstracts were screened independently against the selection criteria by two authors. The full texts were then reviewed and eligible studies were included in the meta-analysis. Any disagreements were resolved by discussion until a consensus was reached or by consulting a third author.

### 2.2. Data Extraction and Outcome Measures

All data were extracted by one author and checked by another author. We also contacted the study authors to request data related to the meta-analysis that was not accessible from the original publications. The following data were extracted: author, year of publication, characteristics of the study population (health condition, sample sizes, age, and gender), study design, rTMS parameters (the number of sessions, stimulation sites, frequency, intensity, total pulses, and timing of stimulation), type of psychological interventions, and pre-intervention and post-intervention outcomes. The outcome measures of interest assessed clinical symptoms and cognitive functions, the former including disease symptom severity (Appendix A) and functional outcomes (Appendix A) measured by standardised questionnaires, and the latter from standardised cognitive tasks or questionnaires (Appendix A). The means, standard deviations (SD), and sample sizes for outcome measures in each group were extracted for the pooled analysis. We also used a Web-based program called WebPlotDigitizer (WebPlotDigitizer, Austin, TX, USA, A. Rohatgi, 2018) to estimate data from figures. If the standard error (SE) was reported, we calculated SD using the equation SD = SE × √N.

### 2.3. Statistical Analysis

Statistical analysis was conducted using R version 4.2.1 (R Core Team, 2022 [37]), RStudio software version 2022.12.0.353 Posit Team, 2022 [38]) and ‘meta’, ‘metafor’ packages. The effect of active or sham rTMS combined with psychological interventions was examined using standardized mean differences (SMD) with 95% confidence intervals (CIs) for each outcome measure as part of a random-effects model. Meta-analyses were conducted when outcomes were available from at least three studies. There were three outcome measures of interest: clinical symptoms, functional outcomes, and cognition. For outcomes where higher scores were associated with poorer performance or more severe disease functional outcomes, these scores were recoded to represent positive effect sizes in favour of the active condition. Cognitive tasks and questionnaires were categorized according to six cognitive domains as defined in the DSM-5, including perceptual-motor function, language, learning and memory, complex attention, social cognition, and executive function [39]. Executive function included tasks which involved updating, shifting, or inhibition [40]. Furthermore, we included two additional cognitive domains: global cognition and working memory. Global cognition was added because a global cognitive score rather than scores within cognitive domains was reported in some of the studies which were included. Similarly, working memory was analysed because it was a commonly assessed cognitive outcome. If multiple cognitive outcomes from the same cognitive task or questionnaire were reported in an individual study, the most commonly used outcome measure as defined by the authors was selected [41] (Appendix A). If the primary outcome measure was not specified for a particular task, we included the most relevant measure based on our predefined cognitive domains. When studies assessed multiple cognitive tasks within the same domain, outcomes from different tasks were averaged to generate domain-specific aggregate effect sizes [14]. Where multiple questionnaires measured the symptom severity for the same disease in each study, the primary outcome measure as defined by the original authors was extracted and analysed [41]. Heterogeneity between studies was assessed using the I^2^ test. I^2^ values > 25%, 50%, and 75% indicate low, moderate, and high heterogeneity, respectively [42]. Secondary subgroup analyses were conducted according to different types of psychological interventions (i.e., rTMS + CBT, rTMS + CT, rTMS + exposure therapy, and rTMS + Mindfulness-Based Stress Reduction (MBSR)), study populations (healthy and clinical populations), and cognitive domains where sufficient data was available for meta-analyses. Given that patients exhibit faster improvement or recovery when subjected to a higher number of psychotherapy sessions [43], we additionally performed subgroup analyses on studies involving 10 or more sessions of combined interventions to examine the effects on clinical symptoms, functional outcomes, and cognition. The revised Cochrane risk of bias assessment tool for RCTs (Risk of Bias tool, RoB 2) was used to independently assess the quality of included studies by two authors. All discrepancies were resolved by consulting a third author. The tool was used to assess bias across five domains: the randomization process, deviations from intended interventions, missing outcome data, measurement of the outcome, and selection of the reported result [44]. Each study was judged as having a low risk of bias, some concerns, or a high risk of bias. Publication bias was assessed using funnel plots and Egger’s test [45] for the outcomes in which 10 or more available studies were included.

## 3. Results

### 3.1. Overview

A total of 3969 articles were identified initially, and 27 studies were ultimately included in the meta-analysis (see Figure 1). The characteristics of the included studies are summarized in Table 1. 

All studies utilized a between-subject design, other than studies from Gy et al. [46] and Osuch et al. [47], which used a within-subject design. In the study conducted by Gy et al. [46], participants were randomized to receive 30 sessions of active or sham rTMS + cognitive stimulation in the first phase and then crossed over to receive the opposite type of rTMS with the same cognitive stimulation in the second phase after a 4-week washout period; we only included the pre- and post-data of the first phase prior to crossover. Osuch et al. [47] reported change scores between baseline and endpoint for each condition, which were then used to calculate the SMD. In studies with multiple treatment arms [48,49,50,51,52,53,54,55,56], only the active/sham rTMS + psychological intervention arms were analysed. Of note, 4 studies had two active arms [50,52,57,58]. In these cases, the sample size of the control group was halved to avoid calculating the same control group twice in the meta-analysis [59]. In summary, 31 treatment arms were included. There were 72 healthy participants and 1060 participants from a clinical population included in the meta-analysis. Clinical populations included patients with low cognitive restructuring ability (*n* = 46), post-traumatic stress disorder (PTSD) (*n* = 180), anxiety disorder (*n* = 83), smokers (*n* = 156), alcohol-dependent patients (*n* = 119), Alzheimer’s disease (AD) (*n* = 167), obsessive compulsive disorder (OCD) (*n* = 30), attention-deficit/hyperactivity disorder (ADHD) (*n* = 62), cognitive impairment (*n* = 88), attention dysfunction (*n* = 58), post-stroke depression (*n* = 47), and major depressive disorder (MDD) (*n* = 24). For healthy participants, data were only available to assess the effect on clinical symptoms and cognition in two RCTs. Four treatment arms involved low-frequency (1 Hz) rTMS, three involved iTBS, and twenty-four involved high-frequency (≥5 Hz) rTMS. Most of the studies (77.8%) stimulated one brain region (especially the dorsolateral prefrontal cortex (DLPFC): 59.3% of total studies), and only six studies stimulated multiple sites. In the included studies, only one utilized a single-session intervention, while the majority employed multiple sessions. Notably, 85.2% of the studies involved 10 or more sessions.

**Table 1 brainsci-13-01665-t001:** Study characteristics.

Intervention	Author (Year)	Subject Sample	Study Design	Sample Size (Active/Sham)	Mean Age ± SD (y) (Active/Sham)	Male Gender (Active/Sham)	rTMS Parameters						Outcomes Measured
							Sessions	Localization	Frequency	Intensity	Total Pulses	Timing of Stimulation	
rTMS + CBT	Neacsiu et al. [58]	Adults with low use of cognitive restructuring	RCT (between-subject)	14/15	33.29 ± 13.98/29.53 ± 10.56	3/2	4	Left DLPFC	10 Hz	120% RMT	800	online	(1) Functional outcome: WSA
				17/15	27.76 ± 7.23/29.53 ± 10.56	3/2	4	Right DLPFC	10 Hz	120% RMT	800	online	
	Hu et al. [52]	Alcohol-dependent patients	RCT (between-subject)	42/37	44 ± 10/46 ± 10	32/28	10	Right DLPFC	10 HZ	110% RMT	1500	offline	(1) Clinical symptoms: OCDS, PHQ-9
				40/37	48 ± 11/46 ± 10	28/28	10	Left DLPFC	10 HZ	110% RMT	1500	offline	
	Kozel et al. [60]	PTSD	RCT (between-subject)	32/30	34.06 ± 7.56/32.93 ± 6.04	31/23	12–15	Right DLPFC	1 Hz	110% RMT	1800	offline	(1) Clinical symptoms: CAPS, QIDS (2) Functional outcome: IPF
	Deppermann et al. [61]	Panic disorder	RCT (between-subject)	22/22	Mean (Range): 37.6 (19–63)/36.3 (22–56)	9/8	15	Left DLPFC	iTBS	80% RMT	600	offline	(1) Clinical symptoms: PAS
	Guhn et al. [62]	Healthy participants	RCT (between-subject)	21/24	23.9 ± 3.0/24.6 ± 4.5	21/22	1	mPFC	10 Hz	110% RMT	1560	offline	(1) Clinical symptoms: PANAS
	Dieler et al. [63]	Smokers	RCT (between-subject)	38/36	23.9 ± 3.0/24.6 ± 4.5	16/24	4	Right DLPFC	iTBS	80% RMT	600	offline	(1) Clinical symptoms: QSU
rTMS + CT	Qin et al. [64]	AD	RCT (between-subject)	10/6	65.60 ± 8.06/66.50 ± 9.40	2/3	20	L-DLPFC and LTL	10 Hz	100% RMT	1000	online	(1) Cognition: MMSE (2) Functional outcome: ADL
	Vecchio et al. [56]	AD	RCT (between-subject)	30/17	71.07 ± 1.25/72.24 ± 2.29	14/10	30	Broca’s area, bilateral DLPFC, Wernicke’s area, bilateral pSAC	10 Hz	Frontal: 90% RMT; other areas:110% RMT	1200–1400	online	(1) Cognition: ADAS-Cog
	Yingli et al. [65]	Post-stroke cognitive impairment	RCT (between-subject)	18/18	60.39 ± 10.87/59.50 ± 11.25	13/12	40	Left or right DLPFC	1 Hz	80% RMT	600	offline	(1) Cognition: LOTCA
	Lechner et al. [54]	Smokers	RCT (between-subject)	12/11	42.50 ±10.45/45.72 ± 9.23	8/8	10	Left DLPFC	10 Hz	100% RMT	2000	offline	(1) Cognition: Maastricht working memory training program, NIH Examiner *n*-back (2 back); NIH Examiner Dot Counting
	Bleich-Cohen et al. [57]	ADHD	RCT (between-subject)	24/16	35.6 ± 8.7/34.7 ± 9.2	17/8	15	Right PFC	18 Hz	120% RMT	1440	online	(1) Clinical symptoms: CAARS, BDI (2) Functional outcome: AAQoL (3) Cognition: Mindstreams, BRIEF-A
				22/16	35.1 ± 10/34.7 ± 9.2	15/8	15	Left PFC	18 Hz	120% RMT	1440	online	
	Gy et al. [46]	Mild cognitive impairment	RCT (within-subject)	22	66.36 ± 5.12	9	30	Left DLPFC	5 Hz	100% RMT	1500	offline	(1) Clinical symptoms: GDS (2) Functional outcome: IWI (3) Cognition: MMSE, MoCA, Stroop, Digit detection, ROCF
	Palaus et al. [66]	Healthy participants	RCT (between-subject)	14/13	29.86 ± 5.26/29.00 ± 7.43	7/6	10	Right DLPFC	iTBS	80%AMT	600	offline	(1) Cognition: Reaction time tasks, 3-Back task, Digit span task, Stop-switching task, Raven’s progressive matrices, Mental rotation task
	Brem et al. [49]	AD	RCT (between-subject)	16/10	69.25 ± 6.80/69.10 ± 5.24	4/5	30	Left IFG, left STG, bilateral DLPFC, bilateral IPL	10 Hz	120% RMT	400	online	(1) Cognition: ADAS-Cog
	Bagattini et al. [67]	AD	RCT (between-subject)	27/23	73.56 ± 4.91/73.53 ± 1.09	17/12	20	Left DLPFC	20 Hz	100% RMT	2000	offline	(1) Clinical symptoms: GDS (2) Cognition: MMSE, Face-name associative memory task, ROCF, RAVLT, phonemic/semantic verbal fluency, Attention matrices, TMT-A, Raven’s progressive matrices
	Li et al. [68]	Post-stroke cognitive impairment	RCT (between-subject)	15/15	65.47 ± 3.68/64.53 ± 4.72	7/9	15	Left DLPFC	5 Hz	100% RMT	2000	offline	(1) Cognition: MMSE, MoCA
	Liu et al. [69]	Stroke patients with attention dysfunction	RCT (between-subject)	29/29	58.55 ± 6.24/57.69 ± 7.25	10/16	20	Left DLPFC	10 Hz	90% RMT	700	offline	(1) Functional outcome: FIM (2) Cognition: MMSE, TMT-A, DST, DS
	Zhang et al. [70]	AD	RCT (between-subject)	15/13	69.00 ± 8.19/68.54 ± 7.93	3/3	20	L-DLPFC and LTL	10 Hz	100% RMT	1000	online	(1) Clinical symptoms: NPI (2) Functional outcome: ADL (3) Cognition: ADAS-cog, ACE-III
	Li et al. [55]	MDD	RCT (between-subject)	12/12	43.4 ± 9.0/39.4 ± 13.2	4/5	10	Left DLPFC	10 Hz	100% RMT	1600	offline	(1) Clinical symptoms/Depression: HDRS-17 (2) Cognition: Visual attention, Go/no-go
rTMS + exposure therapy	Isserles et al. [71]	PTSD	RCT (between-subject)	40/51	44.8 ± 13.19/43.7 ± 12.25	21/21	12	Bilateral mPFC and ACC	18 Hz	100% RMT	2880	offline	(1) Clinical symptoms: CAPS-5, HDRS-21
	Carmi et al. [72]	OCD	RCT (between-subject)	16/14	36 ± 2.1/35 ± 3.5	7/7	25	mPFC and ACC	20 Hz	110% RMT	2000	offline	(1) Clinical symptoms: Y-BOCS
	Herrmann et al. [73]	Height phobia	RCT (between-subject)	20/19	43.2 ± 12.6/46.6 ± 13.7	7/6	2	vmPFC	10 Hz	100% RMT	1560	offline	(1) Clinical symptoms: AQ
	Dinur-Klein et al. [50]	Smokers	RCT (between-subject)	16/15	49.9 ± 12.0/51.6 ± 10.9	11/10	13	Lateral PFC	10 Hz	120% RMT	990	offline	(1) Clinical symptoms: FTND
				7/15	48.3 ± 10.8/51.6 ± 10.9	5/10	13	Lateral PFC	1 Hz	120% RMT	600	offline	
	Isserles et al. [53]	PTSD	RCT (between-subject)	9/9	49 ± 12.5/40.4 ± 10.5	7/8	12	mPFC	20 Hz	120% RMT	1680	offline	(1) Clinical symptoms: CAPS, HDRS-24, BDI
	Osuch et al. [47]	PTSD	RCT (within-subject)	9	41.4 ± 12.3	1	20	Right DLPFC	1 Hz	100% RMT	1800	online	(1) Clinical symptoms: CAPS, HDRS
	Amiaz et al. [48]	Smokers	RCT (between-subject)	12/9	51.5 ± 2.6/48.7 ± 3.5	6/3	10	Left DLPFC	10 Hz	100% RMT	1000	offline	(1) Clinical symptoms: modified FTND
rTMS + MBSR	Duan et al. [51]	Post-stroke Depression	RCT (between-subject)	23/24	58.30 ± 13.06/53.63 ± 13.01	19/20	20	Left DLPFC	10 Hz	80% RMT	1400	offline	(1) Clinical symptoms: HAMD-17 (2) Functional outcome: MBI

Note: AAQOL: Adult ADHD Quality of Life; ACC: anterior cingulate cortex; ACE-III: Addenbrooke’s Cognitive Examination-III; AD: Alzheimer’s disease; ADAS-Cog: Alzheimer’s Disease Assessment scale in cognitive subdomain; ADHD: Attention deficit hyperactivity disorder; ADL: activities of daily living; AMT: active motor threshold; AQ: Acrophobia Questionnaire; BDI: Beck’s Depression Inventory; BRIEF-A: Behaviour Rating Inventory of Executive Function-Adult Version; CAARS: Conners’ Adult ADHD Rating Scale; CAPS: Clinician-Administered PTSD Scale; CBT: cognitive behaviour therapy; CT: cognitive training; DLPFC: dorsolateral prefrontal cortex; DS: Digital Span Test; DST: Digit Symbol Test; FIM: Functional Independence Measure; FTND: Fagerström Test for Nicotine Dependence; GDS: Yesavage’s Geriatric Depression Scale; HAMD: Hamilton Depression Rating Scale; HDRS: Hamilton Depression Rating Scale; IFG: inferior frontal gyrus; IPF: Inventory of Psychosocial Functioning; IPL: inferior parietal lobule; iTBS: intermittent theta-burst stimulation; IWI: interview with informant; LOTCA: loewenstein occupational therapy cognitive assessment; LTL: lateral temporal lobe; MBI: Modified Barthel Index; MBSR: Mindfulness-Based Stress Reduction; MDD: major depressive disorders; MMSE: Mini-Mental State Examination; MoCA: Montreal Cognitive Assessment; mPFC: medial prefrontal cortex; NIH: National Institute of Health; NPI: Neuropsychiatric Inventory; OCD: obsessive compulsive disorder; OCDS: obsessive compulsive drinking scale; PANAS: Positive and Negative Affect Scale; PAS: Panic and Agoraphobia Scale; PFC: prefrontal cortex; PHQ-9: patient health questionnaire-9; pSAC: parietal somatosensory association cortices; PTSD: Post-traumatic stress disorder; QIDS: Quick Inventory of Depressive Symptomatology; QSU: Questionnaire on Smoking Urges; RAVLT, Rey Auditory Verbal Learning Test; RCT: randomised controlled trial; RMT: resting motor threshold; ROCF: Rey–Osterrieth Complex Figure test; STG: superior temporal gyrus; TMT-A: Trail Making Test-A; vmPFC: ventromedial prefrontal cortex; WSAS: Work and Social Adjustment Scale; Y-BOCS: Yale–Brown Obsessive Compulsive Scale.

### 3.2. Clinical Outcomes in Healthy and Clinical Populations

Sixteen studies (19 treatment arms) reported the effect of rTMS combined with psychological interventions on clinical outcomes involving 751 participants: five studies used rTMS + CBT [52,60,61,62,63], three studies used rTMS + CT [55,57,70], one study used rTMS + mindfulness-based stress reduction [51], and seven studies used rTMS + exposure therapy [47,48,50,53,71,72,73]. Of note, data for healthy participants were only available to assess the effect on clinical outcomes in one study using rTMS + CBT [62]. Across the entire sample, significantly greater improvements in clinical symptoms were found after active rTMS + psychological interventions relative to sham rTMS + psychological interventions (SMD  =  0.31, CIs  0.08 to 0.54, Z = 2.66, *p*  <  0.01). A subgroup analysis in clinical populations yielded significance indicating greater improvements in clinical outcomes following rTMS + psychological interventions (SMD  =  0.32, CIs  0.07 to 0.56, Z = 2.54, *p*  =  0.01, I^2^ = 94%). Subgroup analyses according to different types of psychological interventions were further explored. The results showed that neither rTMS + CBT (SMD  =  0.14, CIs −0.03 to 0.32, Z = 1.58, *p*  =  0.11), rTMS + CT (SMD  =  0.59, CIs −0.15 to 1.32, Z = 1.55, *p*  =  0.12), nor rTMS + exposure therapy (SMD  =  0.35, CIs −0.06 to 0.75, Z = 1.66, *p*  =  0.10) had a statistically significant greater benefit for improving clinical outcomes (Figure 2). Moreover, data were only available to do subgroup analyses of rTMS + exposure therapy on clinical symptoms in patients with PTSD and smokers. Results showed no significant difference between active rTMS + exposure therapy and sham rTMS + exposure therapy on clinical symptoms in patients with PTSD (Appendix A) or smokers (Appendix A). Subgroup analysis in studies with 10 or more sessions of combined interventions yielded a significant improvement in clinical symptoms following rTMS + psychological interventions (SMD  =  0.34, CIs  0.07 to 0.61, Z = 2.49, *p*  =  0.01, I^2^ = 94%). Of note, the results showed that the augmentation effects of rTMS became significant for CBT when pooling studies with 10 or more sessions (SMD  =  0.21, CIs  0.05 to 0.36, Z = 2.49, *p*  <  0.01, I^2^ = 78%) (Appendix A). 

A subgroup analysis was additionally conducted to examine the effects of rTMS combined with psychological interventions on depressive symptoms. Ten studies (twelve treatment arms) reported depressive symptom severity outcomes in patients with MDD (*n* = 24), ADHD (*n* = 62), PTSD (*n* = 180), AD (*n* = 50), post-stroke depression (*n* = 47), alcohol-dependent patients (*n* = 119), and mild cognitive impairment (MCI) (*n* = 22) were included. The meta-analysis showed that active rTMS combined with psychological interventions did not produce a significantly greater improvement in depressive symptoms compared to sham rTMS + psychological intervention in the analysed clinical populations (SMD  =  0.06, CIs − 0.16 to 0.29, Z = 0.55, *p*  =  0.58, I^2^ = 86%; Figure 3). A further subgroup analysis in patients with PTSD showed no difference in depressive symptoms between active or sham rTMS when combined with exposure therapy (Appendix A). 

### 3.3. Functional Outcomes in Clinical Populations

Eight studies (ten treatment arms) examined the effect of rTMS + psychological interventions on functional outcomes in clinical populations [46,51,57,58,60,64,69,70]. Meta-analysis revealed that active rTMS + psychological interventions did not significantly improve functional outcomes relative to sham rTMS + psychological interventions (SMD  =  0.10, CIs −0.12 to 0.32, Z = 0.87, *p*  =  0.38; Appendix A). In the subgroup analysis of interventions involving 10 or more sessions, no significant outcomes were observed either (Appendix A).

### 3.4. Cognitive Outcomes in Healthy and Clinical Populations

We investigated the effect of active/sham rTMS + CT on cognitive function from 13 studies (14 treatment arms) for 741 participants across 7 cognitive domains: perceptual-motor function, language, executive function, learning and memory, complex attention, working memory, and global cognition. Data for the effect on cognition were only available in one study in healthy participants (*n* = 27). When data from these ten studies were pooled, there was an overall significantly greater effect with active rTMS + CT compared to sham rTMS + CT (SMD  =  0.28, CIs 0.15 to 0.42, Z = 4.03, *p*  <  0.01). However, the results were highly heterogeneous (I^2^ = 88%). Due to insufficient data, subgroup analyses were limited to the cognitive domains of complex attention, executive function, learning and memory, and global cognition. Results showed that the combination of rTMS + CT had a small effect on global cognition (SMD  =  0.45, CIs 0.21 to 0.68, Z = 3.75, *p*  < 0.01), with significant heterogeneity (I^2^ = 89%). When subgroup analysis in clinical populations was conducted, the results remained significant on overall cognition when all domains were collapsed (SMD  =  0.33, CIs 0.18 to 0.49, Z = 4.22, *p*  < 0.01, I^2^ = 88%) and for global cognition (SMD  =  0.47, CIs 0.22 to 0.72, Z = 3.70, *p*  < 0.01, I^2^ = 89%). Furthermore, the results of subgroup analysis in patients with AD also remained significant for overall cognition (SMD  =  0.43, CIs 0.20 to 0.66, Z = 3.6, *p*  < 0.01; Appendix A) and for global cognition (SMD  =  0.41, CIs 0.08 to 0.75, Z = 2.41, *p*  =  0.02; Appendix A). There was no significant difference between conditions for the complex attention, learning and memory, and executive function domains (Figure 4). 

### 3.5. Publication Bias

The funnel plots for the effect sizes of clinical symptoms, depressive symptoms, functional outcomes, overall cognitive effects, and global cognition showed symmetry, indicating no evidence of publication bias (Appendix A). This was supported by the results of the Egger’s tests (*p* > 0.05). 

### 3.6. Risk of Bias Assessment and Sensitivity Analyses

Two studies (7.4%) were assessed as having a high risk of bias. Herrmann et al. [73] had a high risk of bias arising from the randomization process and Osuch et al. [47] had a high risk of bias due to deviations from the intended intervention (Appendix A). Results from the ROB2 tool are shown in Appendix A. After removing these two studies, the effect of active rTMS combined with psychological interventions was still significantly greater for clinical outcomes compared to sham rTMS combined with psychological interventions (SMD  =  0.32, CIs 0.06 to 0.57, Z = 2.41, *p* = 0.02, I^2^ = 95%). When pooling studies with 10 or more sessions of combined interventions, subgroup analysis showed rTMS had augmentation effects on psychological interventions for overall clinical symptoms with a small-sized effect (SMD  =  0.36, CIs 0.07 to 0.64, Z = 2.47, *p*  <  0.01, I^2^ = 95%) after excluding one study with a high risk of bias [47]. Functional and cognitive outcomes were not affected by the removal of these two studies as they only examined clinical outcomes. 

## 4. Discussion

To the best of our knowledge, this is the first systematic review and meta-analysis to investigate whether rTMS can potentially augment the efficacy of psychological interventions. In summary, the results provided preliminary evidence that: (1) rTMS could potentially augment the effects of psychological interventions on overall clinical symptoms with a small-sized effect across a broad range of health conditions; and (2) active rTMS could potentially produce small-sized cognitive-enhancing benefits on CT for overall cognition in studies involving healthy and clinical populations.

While the combined effects of rTMS and psychological interventions are increasingly being investigated, it remains unclear whether rTMS can augment clinical outcomes. Overall, the results showed that there may be a small-sized beneficial effect of rTMS on psychological interventions for clinical outcomes. More intervention sessions seemed to slightly increase effect size when conducted subgroup analysis including studies with 10 or more sessions of combined interventions. Heterogeneity between studies, however, was high, and subgroup analyses according to different treatment strategies (CBT, CT, and exposure therapy) failed to reach statistical significance, likely due to reduced statistical power available in subgroup analyses. Interestingly, rTMS showed a potential augmentation effect on CBT in studies which included 10 or more sessions, suggesting rTMS may facilitate more rapid clinical effects of CBT with a higher number of sessions. In the subgroup analyses of patients with PTSD or smokers, no beneficial effect was observed following rTMS combined with exposure therapy; however, there were insufficient studies to do subgroup analyses on other health conditions. Of note, only three studies were available for the subgroup analyses of patients with PTSD and smokers, respectively, and the rTMS protocol utilized in each study was unique, which limited statistical power to detect a beneficial effect and so suggests the need for further research. Nevertheless, the current findings provide preliminary and promising evidence suggesting a potential augmentation effect of rTMS on psychological interventions for clinical outcome measures in a broad context. Further research, particularly with larger sample sizes within specific disorders and rTMS protocols with sufficient therapeutic parameters, is warranted to clarify and validate these preliminary findings. 

A recent review provided preliminary evidence that greater improvement in depressive symptoms could be achieved by rTMS combined with psychological interventions, such as CBT and cognitive-emotional reactivation, but no meta-analysis was conducted [3]. Active rTMS combined with psychological interventions was not significantly more beneficial for improving depressive symptoms in clinical populations in this analysis. The result may be due to the large variability in the types of psychological interventions used in this analysis: three studies used rTMS + exposure therapy, four studies used rTMS + CT, two studies used rTMS + CBT, whereas only one study used rTMS + MBSR to investigate the effect on depressive symptoms. A subgroup analysis revealed that patients with PTSD who received rTMS + exposure therapy did not significantly improve their depressive symptoms, which might be partially attributed to no improvement in the clinical symptoms mentioned above. There is a parallel association between improvement in PTSD symptoms and improvements in depression [74]. Depressive symptom outcomes were obtained from different clinical populations, which may also have contributed to the non-significant results. It is also important to note that there were only two studies investigating the effect on depressive symptoms in patients with depression, which limits any conclusions about the use of rTMS to boost the effects of psychological interventions in this population. The majority of patients in the included studies were treated with a minimum of 10 sessions of combined interventions, with either high frequency rTMS applied to the left PFC or low frequency rTMS applied to the right PFC. A large naturalistic study found that rTMS combined with psychotherapy resulted in a 66% response and a 56% remission rate after treatment, with no difference found between high-frequency and low-frequency rTMS [75]. No standard dosage of rTMS was applied, which may also explain the lack of augmentation effects of rTMS on psychological interventions for depressive symptoms. Except for one study, that utilized 2880 pulses per session, the TMS dose ranged from 1400–2000 pulses per session, which was much lower than the FDA-approved rTMS protocol of 3000 pulses per session [75]. Future randomized trials with a larger number of sessions of rTMS and psychological interventions in patients with MDD or comorbid depression are needed to determine the efficacy of this combined intervention for improving depressive symptoms. 

We did not find an additional benefit of rTMS on psychological interventions for functional outcomes in clinical populations or in sub-samples treated with 10 or more sessions. However, it is important to note that only a small number of studies (N = 8) reported functional outcomes, meaning that a lack of effect cannot be concluded from this preliminary analysis, which included diverse functional outcomes measures (e.g., quality of life, vocational and social functioning, and activities of daily living). Deficits in social and vocational functioning, as well as compromised quality of life, are critical and shared features across psychiatric disorders [76,77]. Given the significance of these functional outcomes for patients, it is critical that future research investigates the augmentation effects of rTMS on psychological interventions to assess these outcomes.

Previous meta-analyses have found greater improvement in cognition with active rTMS combined with CT compared to rTMS alone [78,79]. In the present study, we extended these past findings by examining whether active rTMS combined with CT was more beneficial compared to sham rTMS combined with CT in a sample of studies involving healthy and clinical populations. The results showed that overall cognitive functions were potentially improved following active rTMS combined with CT, albeit with a small effect size and significant heterogeneity. Additionally, we also observed a greater improvement in overall cognition following active rTMS combined with CT compared to sham rTMS combined with CT in a subgroup analysis of patients with AD. In addition, high-frequency rTMS applied to the prefrontal cortex, particularly the DLPFC, was used in most of the included studies assessing the effects of rTMS + CT. Existing evidence has demonstrated that high-frequency rTMS targeted over the prefrontal cortex has beneficial effects on cognitive functioning [17,78]. In this study, we provide further evidence that multiple sessions of high-frequency rTMS sessions administered to this region could have additional cognitive-enhancing effects on CT. 

Interestingly, following a subgroup analysis of different cognitive domains, rTMS + CT was only significantly beneficial for improving global cognition with a small effect size. Global cognition measures a wide range of cognitive skills, including orientation, attention, memory, and visuospatial and constructional abilities [80]. A recent systematic review and meta-analysis of neuropsychiatric symptoms across various diagnoses also found rTMS to the left DLPFC alone had a small-sized effect not only for global cognition, but also for declarative memory, working memory, and cognitive control [80]. The current study extends this finding by showing that rTMS might augment the effects of CT in improving global cognition. In this subgroup analysis, nine out of eleven studies examined the effect on global cognition in older populations, who were more likely to be cognitively impaired at baseline. Moreover, a subgroup analysis showed the augmentation effect on global cognition remained significant in patients with AD, who were characterized by significant cognitive impairment and probably benefited most from these combined interventions [81]. The failure to find cognitive-enhancing effects for the executive function, learning and memory, and complex attention domains was likely due to the small number of studies included for these domains. Therefore, we cannot rule out that there are no additional beneficial effects of rTMS on CT for these outcomes at this stage. 

This study has several limitations. First, when a reasonable lower limit of 5 studies is recommended for subgroup analyses [82], the limited number of studies in some subgroups hampers our comprehensive assessment of the effects of certain treatment strategies (e.g., rTMS + MBSR), outcome measures, or in certain disorders (e.g., panic disorders, ADHD). This limited statistical power to detect potential beneficial effects of rTMS on psychological interventions, suggesting that more studies across diverse categories are warranted to validate and further extend upon these preliminary findings. Secondly, the included studies exhibited substantial clinical and methodological heterogeneity. We attempted to conduct subgroup and sensitivity analyses to identify potential sources of heterogeneity. However, the heterogeneity remained high. The current study examined the effects of four different treatment strategies (various combinations of rTMS and psychological intervention parameters) in participants with twelve health conditions on three outcome measures grouped into different domains. Thus, aggregating effects from different conditions likely contributed to heterogeneity. The high heterogeneity made it challenging to draw meaningful and generalizable conclusions from our current meta-analysis. Future research would benefit from including larger sample sizes in different clinical populations and further examination of the specific efficacy of rTMS with a standardized protocol (e.g., sufficient number of sessions) on psychological interventions in improving clinical, functional, and cognitive outcomes. Third, the combination of rTMS with psychological interventions is a relatively new and emerging area of research, so the number of included studies in the respective analyses was limited, which limited the statistical power to detect bias. Fourth, a high proportion of included studies were assessed as having some concern of risk of bias, while two studies exhibited high risk. Nevertheless, sensitivity analyses showed that the results were not changed on overall cognition after the removal of these two studies. Fifth, we only included studies comparing active or sham rTMS combined with active psychological interventions. Future research involving sham psychological interventions is needed to determine whether improved clinical outcomes with rTMS combined with psychological interventions are due to synergistic or additive mechanisms. 

## 5. Conclusions

In conclusion, this study provides preliminary evidence that active rTMS may augment clinical outcomes when combined with psychological interventions. Further, rTMS might produce a small-sized cognitive-enhancing effect on CT, warranting further research into this combined intervention. These preliminary results, however, must be interpreted with caution due to high heterogeneity and, therefore, require confirmation and replication in future research. In particular, studies with large sample sizes are needed to better understand and optimize the combined effect of rTMS and psychological interventions for specific clinical conditions. 

## Figures and Tables

**Figure 1 brainsci-13-01665-f001:**
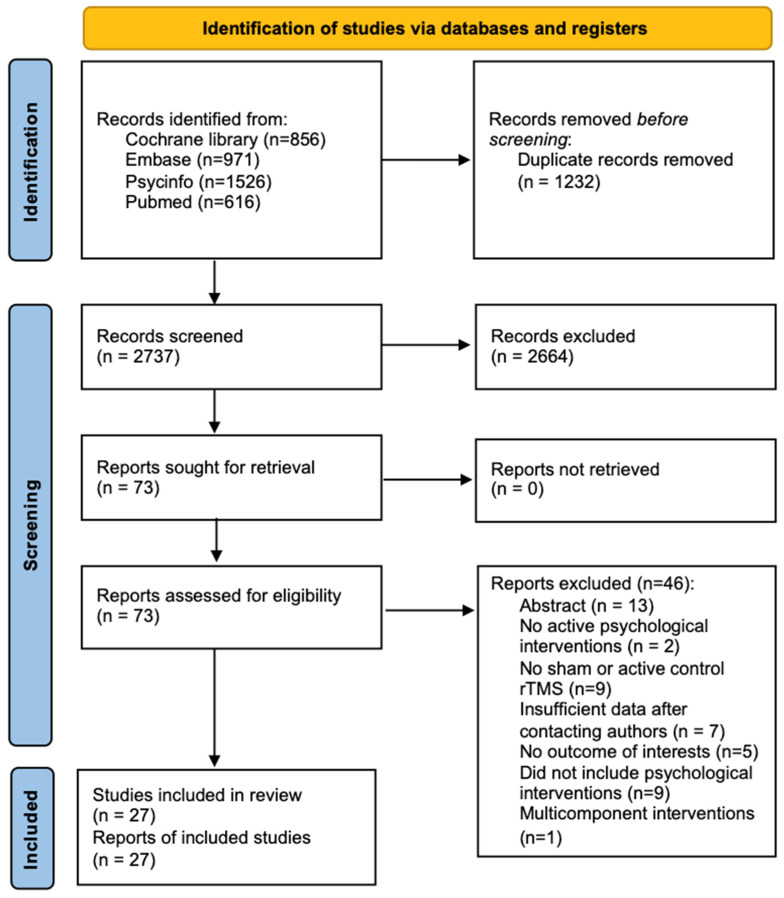
Flowchart of study searching and selection.

**Figure 2 brainsci-13-01665-f002:**
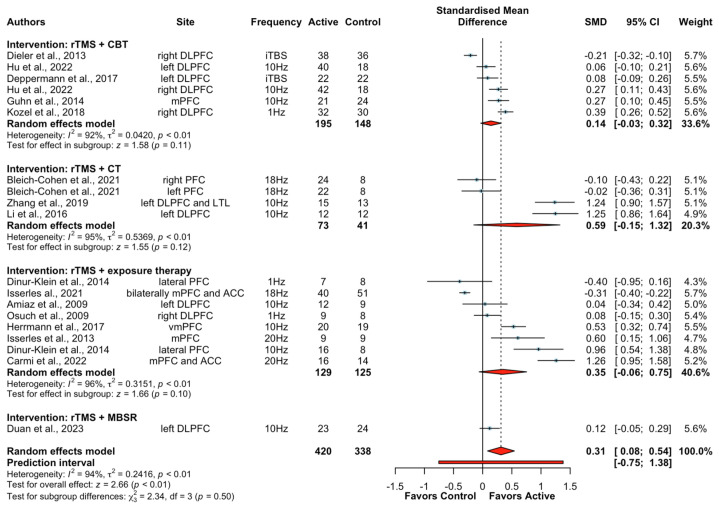
Forest plots of the effect of rTMS + psychological interventions on clinical symptoms [47,48,50,51,52,53,55,57,60,61,62,63,70,71,72,73]. Note: CI: confidence interval; SMD: standardized mean differences.

**Figure 3 brainsci-13-01665-f003:**
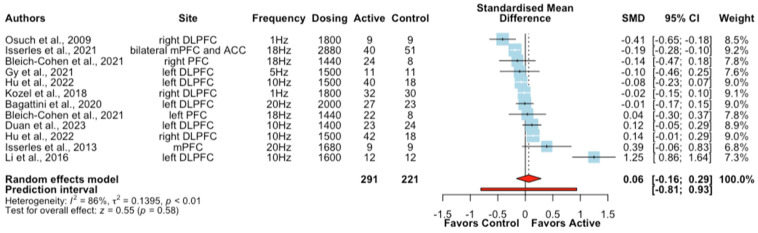
Forest plots of the effect of rTMS + psychological interventions on depressive symptoms [46,47,51,52,53,55,57,60,67,71].

**Figure 4 brainsci-13-01665-f004:**
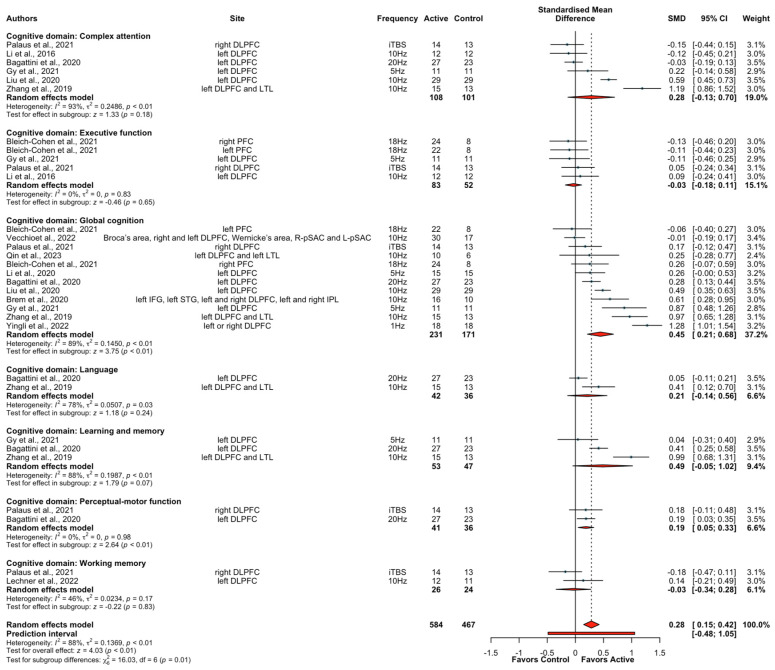
Forest plots of the effect of rTMS combined with cognitive training for different cognitive domains [46,49,54,55,56,57,64,65,66,67,68,69,70].

## Data Availability

All relevant data is contained within the article: The original contributions presented in the study are included in the article/Appendix A, further inquiries can be directed to the corresponding author.

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
