# Peer review of "Efficacy of Repetitive Transcranial Magnetic Stimulation (rTMS) Combined with Psychological Interventions: A Systematic Review and Meta-Analysis of Randomized Controlled Trials"

_brainsci, 2023, doi:10.3390/brainsci13121665_

Round 1
Reviewer 1 Report
Comments and Suggestions for Authors
Efficacy of repetitive transcranial magnetic stimulation (rTMS) combined with psychological interventions: A systematic review and meta-analysis of randomized controlled trials
Thank you to the authors for their effort in preparing this challenging meta analysis. The authors address a worthy and timely question regarding whether or not the addition of rTMS to psychological interventions has greater efficacy than sham rTMS and psychological intervention.
Potential major limitations based on the introduction of the paper:
I believe the authors did as good a job as possible given the complexity of this literature, however there are some major limitations with some aspects of this research. Unfortunately, I think some aspects of the paper may be misleading to those who do not read it in acute detail. The very premise of the paper is about rTMS+-psychological intervention vs sham rTMS+psychological intervention. When someone reads this (scientist, clinician, rTMS provider, patient) you would expect that any real world intervention would at minimum include a sufficient number of rTMS sessions (20+) and therapy sessions (eg. 10+). But the literature and this meta analysis included many papers with only a few sessions that would not be expected to have a meaningful clinical benefit.
I also think the authors need to make it explicitly clear in the introduction that the premise is to use rTMS to augment psychological interventions and not the other way around, because the set up of the paper begs for there to be conditions of rTMS+psychological intervention vs only rTMS or rTMS+ some kind of sham therapy. This meta analysis also tries to impossibly compare across many different pathologies for which rTMS is not yet completely established.
Potential major limitations based on the conclusion of the paper:
I admire the authors effort for conducting this rather complex meta analysis. Given the complexity of the meta analysis though I do have some fundamental issues with the methodology. It doesn't make sense to me to or feel valid to attempt to draw conclusions (positive or negative findings) from a such a heterogenous array of studies. Many of the studies did not include a sufficient number of rTMS sessions that would be expected to have a significant effect on clinical symptoms, functioning, or cognition. I think that the report of the meta analysis is useful from the point of view of the systematic review and reporting of the work, but I don't think any real meaningful conclusions can be made from this at all - and any conclusions that are being presented may appear misleading. To draw more meaningful conclusions it would be critical for the authors to conduct additional sub group analysis specific to each disease using only clinically sufficient rTMS paramters (eg. 20+ number of rTMS sessions). Moreover, even the authors own conclusions are confusing as you state that overall there is some effect of psychological intervention with rTMS but in every sub analysis you failed to find a meaningful effect... and the effect you did find cannot be meaningfully interpreted because it applies to several different forms of therapy across several different pathologies. I think the authors need to really spend more time addressing these poitns in their limitation section.
Minor points:
line 130 How are defining the most commonly use outcome measure? Is this most common among the papers reviewed in the present review, or in the literature in general? The former would be easier to define, the latter may be impossible to definitively define? - the authors do a nice job of defining questionnaire outcomes in the following sentence, they should do similar here regarding cognitive tasks
line 311 - current conventions suggest using person first language -- eg. saying "patient's with PTSD", not "PTSD patients"
line 325 - I'm a believer in the potential for tDCS but this sentence seems misplaced. This sentence should be explored further (elaborate), or be deleted / moved elsewhere.
Despite all of the above critiques I do think this paper is still a useful reference for the literature, however the authors should be very clear about exactly what can and cannot be concluded from this paper. I think the paper would feel a lot more honest to the reader if the authors addressed the paper as I am here …. A useful reference filled with a good summary of literature and statistics that still remains extremely heterogenous and therefore near impossible to draw meaningful conclusions from. I think the authors did one or a few additional sub group analysis using only well established rTMS parameters they could at least draw a valid conclusion based on the analysis.
Reviewer 2 Report
Comments and Suggestions for Authors
In summary, this systematic review and meta-analysis reports on the combined effects of repetitive transcranial magnetic stimulation (rTMS) and psychological interventions for neuropsychiatric symptoms. They found that active rTMS, when combined with psychological interventions, significantly improved clinical symptoms compared to sham rTMS. Additionally, rTMS combined with cognitive training improved overall cognition. In conclusion, combining rTMS with psychological interventions may have small but positive effects on clinical outcomes and cognition.
I would like to make a series of improvement suggestions to the authors:
Heterogeneity Management: The study acknowledges high heterogeneity across included studies. To enhance scientific robustness, efforts should be made to explain the sources of this heterogeneity. Further exploration and discussion of this aspect can help readers understand the variations in the outcomes.
Sample Diversity: The study includes a variety of health conditions, which can impact the interpretation of results. To improve the scientific quality, a more detailed discussion about how different health conditions might influence the combined effects of rTMS and psychological interventions is needed.
Subgroup Analyses: While the study conducts subgroup analyses, the limited number of studies in some subgroups may affect the reliability of the findings. A more extensive discussion on the statistical power and the implications of subgroup analyses with a small number of studies is necessary.
Depressive Symptoms: The non-significant effect on depressive symptoms, especially in patients with depression, requires a more in-depth discussion. The potential reasons for this outcome and its implications for clinical practice should be thoroughly explored.
Functional Outcomes: The study's finding of no additional benefit on functional outcomes is a critical aspect that deserves further discussion. Clarification regarding the importance of functional outcomes in psychiatric disorders and potential reasons for the observed results is necessary.
Publication Bias: The assessment of publication bias is mentioned but not detailed in the "Results" section. A more comprehensive explanation and discussion of publication bias for the key outcomes can improve the scientific quality of the study.
Limitations: The limitations of the study are mentioned, but they could be more extensively discussed, especially in terms of the potential sources of bias and how they might impact the findings. Addressing these limitations would enhance the transparency of the research.
Future Research: While the study touches on the need for further research, a more thorough exploration of specific areas for future investigation would provide a more comprehensive conclusion and guide the direction of future studies.
Synthesis of Results: The findings from different sections of the article (clinical, functional, and cognitive outcomes) could be better synthesized in the discussion to provide an integrated perspective on the combined effects of rTMS and psychological interventions.
Final Remarks: The conclusion could be more comprehensive, summarizing the key findings and their implications for clinical practice, research, and patient care.
With these suggestions in mind, you can revise the article to make it clearer and more accessible to the average reader. Feel free to modify them as needed based on the specific requirements of the research article. Overall, the article is very good.
I want to express my sincere appreciation for the time and effort that you have invested in your research.
It has been a pleasure reviewing your work and I am confident that with the suggested revisions, your paper will make a valuable contribution to the field. I wish you all the best in your continued research.
Best regards,
Round 2
Reviewer 1 Report
Comments and Suggestions for Authors
Thank you to the authors for their time in addressing my earlier comments. I feel the authors have sufficiently addressed my earlier critiques. I think there are still some inherent limitations when pooling all of these pathologies and protocols into a single paper but the authors revisions pertaining to these limitations is upfront and gives the reader pause to interpret findings carefully. The inclusion of the sub group analysis of 10 or more sessions added value to the paper. I agree further research and analyses will be necessary to draw more meaningful conclusions specific to each unique pathology and protocol. This paper is still a useful reference for readers aiming to combine or augment therapy/TMS.